# Magnetosheath jets at Jupiter and across the solar system

Yufei Zhou [1], Savvas Raptis [2], Shan Wang [3], Chao Shen [1] ✉, Nian Ren[1,4] & Lan Ma[1]

The study of jets in the Earth's magnetosheath has been a subject of extensive investigation for over a decade due to their profound impact on the geomagnetic environment and their close connection with shock dynamics. While the variability of the solar wind and its interaction with Earth's magnetosphere provide valuable insights into jets across a range of parameters, a broader parameter space can be explored by examining the magnetosheath of other planets. Here we report the existence of anti-sunward and sunward jets in the Jovian magnetosheath and show their close association with magnetic discontinuities. The anti-sunward jets are possibly generated by a shock–discontinuity interaction. Finally, through a comparative analysis of jets observed at Earth, Mars, and Jupiter, we show that the size of jets scales with the size of bow shock.

Collisionless shocks are abundant in space, manifesting as bow shocks in front of planets, comets and asteroids. Among them, the Earth's bow shock has received the most extensive investigation given its proximity to our planet and the ability to measure it through in-situ measurements. Recent studies have shed light on a crucial characteristic of the bow shock: the occurrence of jets in its downstream region (see, e.g.[1–5]). Jets are transient enhancements in plasma dynamic pressure which typically surpass the dynamic pressure of the upstream solar wind[6]. Consequently, jets can have strong impacts on their relevant environments. In the Earth's context, they have been suggested to indent the magnetopause over a large spatial scale and thus driving a sunward flow[7], exciting eigenmode waves[8] or triggering magnetic reconnection on the magnetopause[9], accelerating electrons in the magnetosheath[10], or driving ultra-low frequency magnetic waves on the ground[11].

Most jets are observed in quasi-parallel magnetosheath, which is the shocked plasma originating from a quasi-parallel shock crossing. Several mechanisms have been proposed to account for the formation of jets at quasi-parallel geometry, with the majority of them being linked to the robust presence of foreshock in quasi-parallel shock scenarios. These mechanisms include the generation of jets through the inherent shock rippling[12], shock reformation[4,13], and shock interaction with rotational discontinuities[14]. Quasi-perpendicular shocks are considered unfavorable for jet formation. Since the sub-solar bow shocks of outer planets are more frequently quasi-perpendicular due to the Parker spiral, it is questioned whether or not jets exist downstream of these shocks. Recent studies suggest that non-shock processes and structures in quasi-perpendicular magnetosheath[15] and shock interaction with discontinuities at quasi-perpendicular shocks[16] can also be responsible for the formation of jets, thus implying that jets may also develop in other planetary magnetosheath.

Although the properties and possible origins of jets downstream of the terrestrial bow shock have long been investigated, their parametric variation with solar wind, obstacle (e.g. magnetosphere), and shock conditions has not been studied nor have their effects on planetary systems and their evolution from parametric shock dynamics been appreciated. Until recently, only one report had been made on the clear presence of jets outside of Earth's magnetosphere, specifically in the magnetosheath of Mars[17], although previously, isolated magnetic field structures were observed at Mercury that could potentially be magnetosheath jets[18].

Shock acceleration is a fundamental source of energetic particles throughout the universe, yet our understanding is still insufficient, and

[1]School of Science, Harbin Institute of Technology (Shenzhen), Shenzhen, China. [2]Applied Physics Laboratory, Johns Hopkins University, Laurel, MD, USA. [3]Institute of Space Physics and Applied Technology, Peking University, Beijing, China. [4]School of Physics and Electronic Science, Hunan Institute of Science and Technology, Yueyang, China. ✉ e-mail: shenchao@hit.edu.cn

many associated phenomena are still, to a large extent, unexplored. Magnetosheath jets have been shown to accelerate particles[19,20], which indicate that transient structures both upstream[21,22] and downstream of shocks that have been neglected in shock acceleration theories can result in additional acceleration, providing seed populations and enhancing shock acceleration efficiency. This scenario is particularly pronounced for discontinuity-driven jets because discontinuities have been shown to drive both foreshock transients and magnetosheath jets[14,16]. Observing them across different planets could help establish the applicability of the scenario in general shock environments, from planetary ones to high Mach number astrophysical shocks.

In this study, we present new findings of anti-sunward and sunward jets downstream of the bow shock of Jupiter and show that some of the jets are possibly related to shock–discontinuity interactions. Furthermore, by incorporating the data from other studies of Martian[17] and terrestrial jets[7,23,24], we investigate the parametric variation of jets in relation to the conditions of each planetary bow shock. Finally, as a preliminary result, we also incorporate a possible jet at Mercury[18] and a tentative jet from Kronian magnetosheath into the comparative study.

## Results

### Voyager 2 observation

At Jupiter, the Voyager 2 spacecraft remains the sole probe to have traversed the subsolar region of the Jovian magnetosheath[25]. It crossed the Jovian bow shock at 17:35 UTC on 1979-07-03, entered the magnetosphere temporally around 00:00 on 1979-07-05 and eventually at 18:40 on 1979-07-05[26]. Figure 1d–k show data from Voyager 2 from 17:00 on 1979-07-03 to 20:00 on 1979-07-04. Figure 1d–e provide the magnetic field in Jupiter-Sun-Orbital (JSO) Cartesian and spherical coordinates. In this coordinate system, the $x$-axis points from Jupiter to the Sun, the $z$-axis is aligned with $\mathbf{v}_{JO} \times \hat{x}$ (where $\mathbf{v}_{JO}$ denotes the orbital velocity of Jupiter), and the $y$-axis completes the right-handed system. Plasma data are presented in Fig. 1h–k. The ions in the Jovian magnetosheth have two-temperature proton distributions[27]. Thus the distributions were fitted using two Maxwellians in the released data product, resulting in a cold ion component (Fig. 1i) and a hot one (Fig. 1j), both having the same bulk velocity. For the duration of observation, Fig. 1a–c show the trajectory of Voyager 2 (blue arrow) as it moved toward Jupiter[28,29]. The three panels represent projections

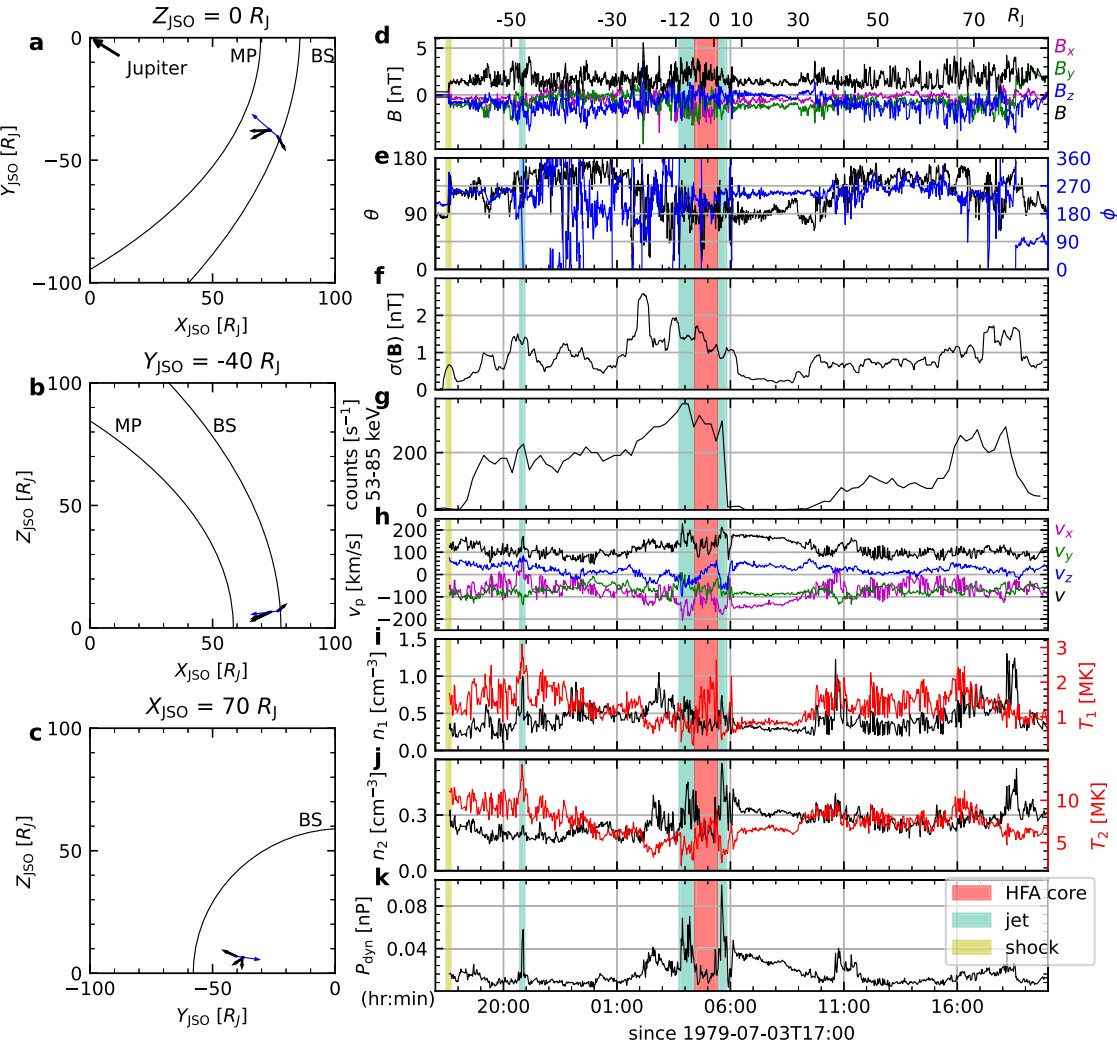

**Fig. 1 | Jovian magnetosheath observation.** Left: trajectories of Voyager 2 (blue arrow) projected onto three planes of the JSO (Jupiter-Sun-Orbit) coordinate system: **a** the $Z = 0$ plane, **b** the $Y = -40R_J$ plane, and **c** the $x = 70R_J$ plane. The positions of bow shock and magentopause are represented by two black curves. Each black arrow attached to the trajectory of Voyager 2 shows the direction and magnitude (proportional to the arrow length) of velocity of a jet observed at the location. Right: Color shadings mark different plasma regions. The displayed quantities are magnetic field in JSO **d** Cartesian coordinates and **e** spherical coordinates; **f** running standard deviation of magnetic field; **g** 53–85 keV ion counts per second; **h** ion bulk velocity; number density and temperature of **i** cold ion component and the **j** hot ion component; **k** ion dynamic pressure. The scales at the top of panel **d** indicate the distance traveled by the spacecraft within the magnetosheath flow. Source data are provided as a Source Data file.

onto the plane of $Z_{JSO} = 0$, $Y_{JSO} = -40R_J$, and $X_{JSO} = 70R_J$, respectively. Three jets (turquoise-shaded areas) are identified from the observed dynamic pressure (Fig. 1k). Their velocities and locations of encounter are marked by black arrows in Fig. 1a–c.

## Sunward jet

The first jet was moving sunward with a positive velocity along $x$-direction (Fig. 1h). It exhibited notable increases in both speed and density, contributing to its significantly higher dynamic pressure compared to the relatively calm background flow. The magnetic field in the jet dropped to a lower level than the ambient in magnitude and showed a rotation in its direction. Additional features associated with the jet were temperature increases of both cold and hot ion components, a pulsing peak in the cold ion density, and a mild increase in the hot ion density.

Sunward flows have also been reported using data from Voyager 1 during its flyby of Jupiter and THEMIS data at Earth[7,23,30]. These reported flows were observed near magnetopause crossings, leading to suggestions that a global expansion of the magnetopause due to a decline in solar wind dynamic pressure or a rebounding of the local magnetopause after being impacted by an anti-sunward jet could drive a sunward flow[13]. The jet presented here was near a crossing of bow shock and far from the subsequent magnetopause crossing, which suggests that it may instead be associated with other processes occurring in the magnetosheath or with the bow shock. A possible explanation could be the presence of a hot flow anomaly for which simulations have shown a flow reversal in their core hot flow anomaly (HFA)[31]. However, the density in the sunward flow was higher than the ambient flow which was in contradiction with density-depleted HFA. Thus, HFA cannot explain the jet.

The Jovian magnetosphere is more compressible than the terrestrial one[28], indicating that the magnetosphere contract and expand more easily in response to a change in solar wind dynamic pressure. Since it takes hours for plasma to travel the length of the Jovian dayside magnetosheath ($\sim 20R_J$, see the scale on top of Fig. 1d), the breathing, i.e. the contraction and expansion, of the magnetopause is expected to occur on a time scale of hours. Indeed, the breathing effects are visible in the variation of $v_x$ in magnetosheath (Fig. 1h). The increasing (decreasing) $v_x$ over hours indicates an expanding (contracting) magnetosphere. Therefore, the sunward jet appears within the expansion phase of the magnetosphere. However, while the breathing magnetosphere provides a context, it alone cannot account for the formation of the jet, as it does not explain the observed increase in speed, density, and temperature within the jet.

## Anti-sunward Jets

The second and third jets occurred within a close time frame of approximately two hours, coinciding with the duration of a HFA observed upstream of the Jovian bow shock by the Juno spacecraft[32]. Between these two jets, there was a decrease in density and an increase in temperature for both ion components (Fig. 1i, j). A rotation of magnetic field was also observed. These features were reminiscent of the cores of HFA convected downstream, as observed in Earth's magentosheath[16,33,34]. The magnetic field in this region had an overall increase, except a localized decrease. This magnetic feature is also shown by the previously identified upstream HFA at Jupiter[32]. The magnetic field and plasma properties before the second jet exhibited significant variations, indicating quasi-parallel magnetosheath conditions. After the third jet, the spacecraft was passed by a quiet flow region in which the average direction of magnetic field vector was $[\theta = 90°, \phi = 245°]$, which in conjunction with the location of observation shown in Fig. 1a suggested that this region was a quasi-perpendicular magnetosheath. To further support these identifications, we utilize magnetic field standard deviation (see Methods subsection standard deviation of magnetic field) and high energy ion

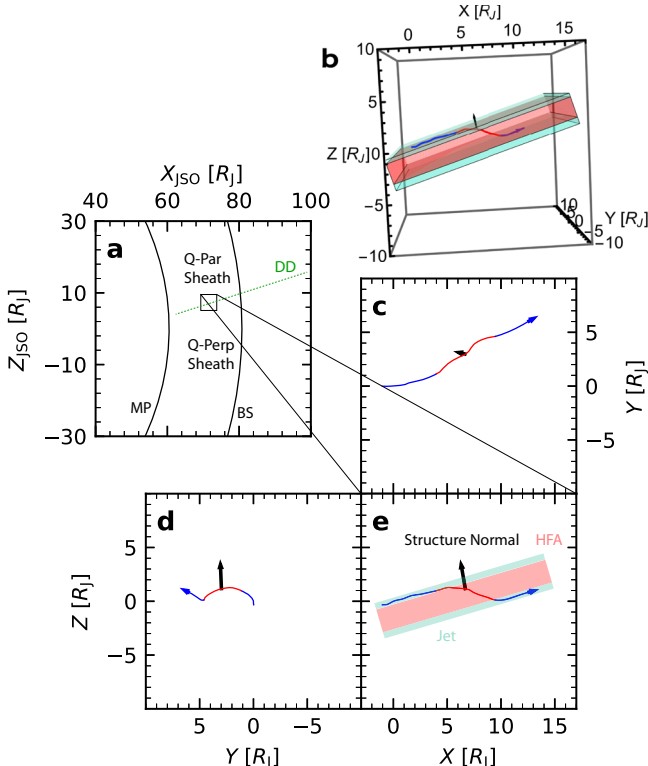

**Fig. 2 | Schematic of the jets and hot flow anomaly (HFA) encountered by Voyager 2. a** A (green) directional discontinuity (DD) sweeping through the bow shock (BS) and possibly the magnetopause (MP), downstream of which was quasi-parallel (Q-Par) magnetosheath and upstream of which was quasi-perpendicular (Q-Perp) magnetosheath. **b** The 3-dimensional view of the trajectory of Voyager 2 through the (cyan) jets and (red) HFA. The spacecraft trajectory projected onto the **c** $Z = 0$, **d** $X = 0$, and **e** $Y = 0$ plane, with blue (red) color representing that it was in a jet (HFA). The black arrow represents the normal of the HFA. Source data are provided as a Source Data file.

flux (Fig. 1f–h), which have been suggested as good local indicators to distinguish between quasi-parallel and quasi-perpendicular magnetosheath[6,35]. The distinction between the two regions were obvious. The region before the two jets were characterized by higher magnetic variance and higher high energy ion flux, while the region after them were of low values of the two indicators. The rotation of the magnetic field vector across the two jets, along with the presence of a heated and density-depleted region between them, further supported the interpretation that this could be a downstream HFA.

Figure 2b shows the trajectory of Voyager 2 in the plasma flow (see Methods subsection Voyager 2 trajectory in plasma and jet size), assuming the overall structure was stable during the spacecraft traversal. In Fig. 2c, d, three different views of the trajectory are presented. The black arrow indicates the direction of minimum magnetic variance of the HFA core, determined using the MVAB method[36] applied to the magnetic field data from 04:23:12 (h:min:s) to 05:28:12 (An alternative method to determine the direction is given in Supplementary Methods). The color shadings in the figures represent a schematic representation of the jets and the HFA, based on the spacecraft trajectory and the normal direction. This schematic view bears similarities to a previous simulation on shock interaction with discontinuities[31] (Fig. 2 of the paper). A possible account for the two jets is as follows: The two density peaks contributing to the dynamic pressure corresponded to the plasma pileups on the leading and trailing edges of the HFA. The increased velocity in the jets may be attributed to the curved shock formed during shock–discontinuity interaction, since curved shocks are less capable of decelerating

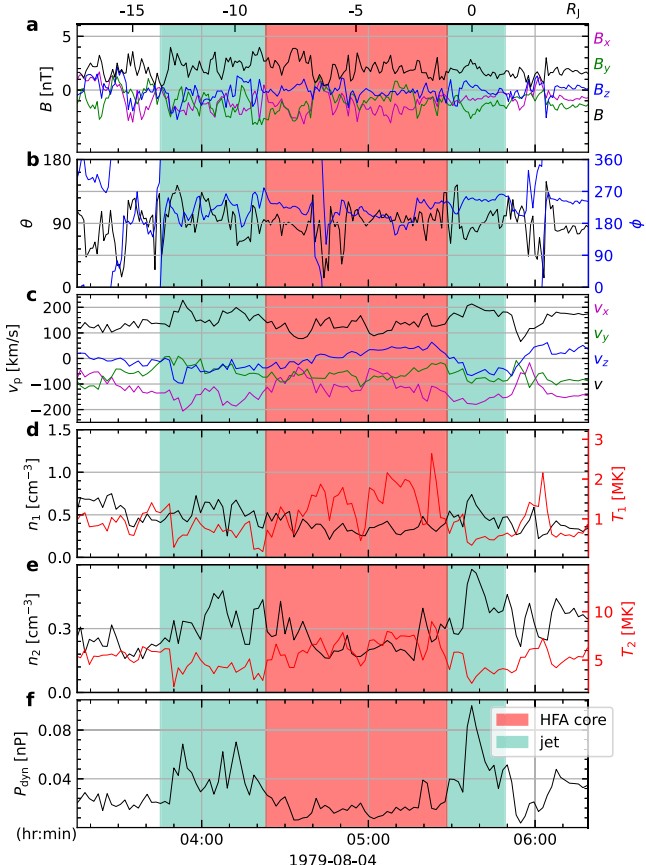

**Fig. 3 | Zoomed in view of the second and third jets shown in Fig. 1.** The displayed quantities are magnetic field in JSO (Jupiter-Sun-Orbit) **a** Cartesian coordinates and **b** spherical coordinates; **c** ion bulk velocity; number density and temperature of the **d** cold ion component and the **e** hot ion component; **f** ion dynamic pressure. The scales at the top of **a** indicate the distance traveled by the spacecraft within the magnetosheath flow. Source data are provided as a Source Data file.

flows and thus could produce relatively high speed flows than other shocks[16,31,37].

Other than shock–discontinuity interactions, observations at Earth suggest that mirror modes, commonly found in quasi-perpendicular magnetosheath, can also contribute to the generation of jets[15,38,39]. According to a previous statistical study[40], in the Jovian magnetosheath mirror modes were found during 62% of the time when data were available; and indeed, mirror structures had been identified around the two jets by using magnetic field data only. Nonetheless, mirror structures within the Jovian magnetosheath typically exhibit durations ranging from 20 to 120 seconds[40,41] and sizes equivalent to 20 proton gyroradii, approximately ~ $0.15 R_J$[41,42], thus not likely being responsible for jets reported here. The positively correlated magnetic field and density in the third jet as seen in Fig. 3 also denies mirror mode at this meso-scale. Instead, they suggest a fast mode nature, which is consistent with the interpretation of being the boundary of a HFA. The correlation in the second jet is less clear.

**Comparing magnetosheath jets across planets**

The jets in the Jovian magnetosheath had a long duration in spacecraft observation, which exceeded ten minutes and can even approach an hour. This is much longer than the durations observed for jets at Earth (up to ~ 3 min)[43]. By using the measured ion speeds and the jet duration, the parallel size of the jet along its flow direction can be estimated (see Methods subsection Voyager 2 trajectory in plasma and jet size).

Here we compare jets at Mars, Earth, and Jupiter. We include four jets at Mars that were recently reported using MAVEN observation[17], where they were grouped as two jets given the proximity of the jets in each pair. One such pair is depicted in Fig. 4. In this study, we separate them as they show similar features to the second and third jets observed at Jupiter. For terrestrial jets, we include two sunward jets reported in[7,23] and two anti-sunward jets reported in[7,24]. Figure 5 shows the parallel sizes of these jets plotted against the bow shock standoff distance. The parallel size of jets appears to scale with the size of bow shock. A similar trend scaling with shock size has also been found for HFA upstream of planetary bow shocks[32,44]. Figure 5 indicates that jet size is proportional to the square root of the bow shock standoff distance. However, it is important to acknowledge that this result is limited by the relatively small number of observed jets in planetary magnetosheaths so far.

In the Supplementary Information, we present a preliminary case report of a sunward jet in the Kronian magnetosheath (see Supplementary Fig. 1). Additionally, we integrate this Kronian jet as well as a previously documented possible jet at Mercury[18] into our comparative analysis. Due to the limitation of plasma measurement at Mercury, it is difficult to include the Mercurian possible jet into the comparison of size. Since the flow speed in these planetary magnetosheath are of similar magnitude, a comparison of jet duration is presented to show the same scaling trend with bow shock size for all these planets (see Supplementary Fig. 2).

In the various mechanisms accounting for their origin[4,12–16], jets have been associated with the microphysics of shock waves. Thus, it is possible that the size of jets scale with ion inertial and cyclotron length which, according to the Parker theory of solar wind, increase roughly linearly with the solarcentric distance beyond the orbit of Mars. This possible trend, however, is not shown in the jets observed so far.

## Discussion

In summary, we have reported anti-sunward and sunward jets in the Jovian magnetosheath. These jets were associated with magnetic field discontinuities. The pair of anti-sunward jets at Jupiter likely resulted from shock interaction with solar wind discontinuities and coincided with the pileup regions at the edges of a HFA. The origin of the sunward jets remains an open question. Combining all these observations along with recent ones made at Martian magnetosheath as well as terrestrial observations, we have shown that the parallel size of jets scales with the size of bow shocks, indicating a mesoscale nature of jet formation and evolution.

Early studies on jets at Earth[1] led to the prevailing notion that jets could only form at quasi-parallel shocks where rippling geometry naturally occurs, though recent studies suggest that jets can also be associated with structures in the quasi-perpendicular magnetosheath[15]. Questions arose about whether a non-trivial number of jets could emerge from the subsolar Jovian and Kronian bow shocks, which are predominantly quasi-perpendicular such that ion foreshock does not prevail (the mean interplanetary magnetic field cone angles are 79° and 84° respectively according to Parker spiral). This study provide observational evidence that although Parker spiral is not ideal for subsolar foreshock at Jupiter and Saturn, the highly intermittent and discontinuous magnetic fields in solar wind allow jets to be formed downstream of these planetary bow shocks[16,45–47].

Shock–discontinuity interaction is an explosive process closely associated with shock dynamics[48]. It can occur at both quasi-parallel and quasi-perpendicular geometries[49–51]. A standard picture of the interaction has been gradually established over the past four decades[52–54], with the recent addition of its effect on acceleration of ions, in which first-order Fermi acceleration is set on when the newly formed shock at the HFA compressive boundary moves relative to the planetary bow shock[22]. Since previous studies suggested jets can result from shock–discontinuity interactions[16,55,56] and this study provide

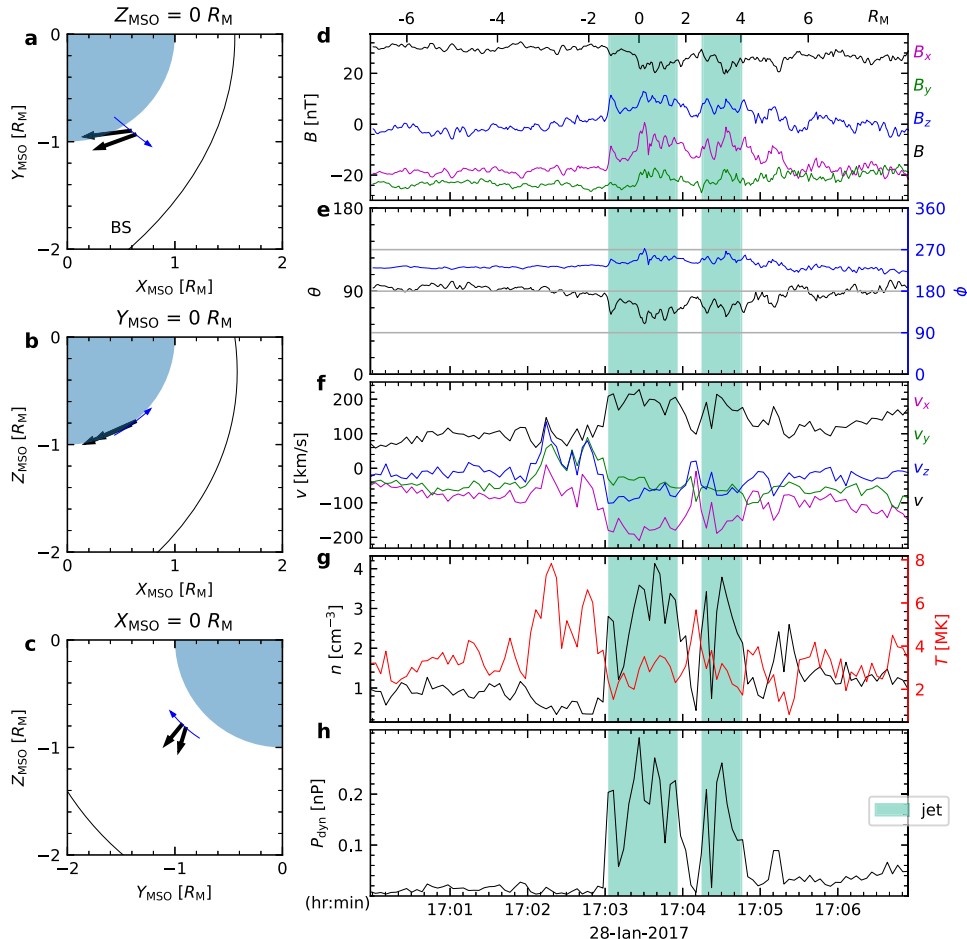

**Fig. 4 | Jet observations in Martian magnetosheath.** Same format as Fig. 1. Left : **a**–**c** spacecraft trajectory plots in the MSO (Mars Solar Orbital) coordinates. Bow shock is modeled using[70]. Right: Color shadings mark jets. The displayed quantities are magnetic field in MSO **d** Cartesian coordinates and **e** spherical coordinates; **f** ion bulk velocity; **g** ion number density and temperature; **h** ion dynamic pressure. The scales at the top of panel **d** indicate the distance traveled by the spacecraft within the magnetosheath flow. Source data are provided as a Source Data file.

further support, it is of interest to incorporate the generation of jets into the standard picture and study the role they play in the overall process (e.g. how they affect the geometry in the evolution of HFA and hence affect the acceleration process). Figure 1g shows an enhancement of energetic ion flux starting before the second jet and peaking within the jet. This may be associated with acceleration in the upstream HFA[22] or with acceleration by a secondary bow wave or shock driven by the jet[19]. To explain this enhancement is out of the scope of this study.

The Jovian and Kronian bow shocks exhibit high Mach numbers (Alfvèn Mach number $M_A \sim 10$) and, in rare cases, can even reach very high Mach numbers ($M_A \sim 100$)[57]. The frequent quasi-perpendicular geometry of their subsolar sections, combined with high Mach numbers, makes them ideal candidates for in-situ studies of shock conditions that are common in astrophysical shocks[58]. The observational result of jet in this study could potentially be applied to those distant astrophysical shocks.

Theoretical calculations suggest that the strength of jets (ratio of jet to upstream dynamic pressure) grow with Mach number and decrease with plasma beta (ratio of thermal to magnetic pressure)[59]. Given the high Mach number for Jovian and Kronian bow shock and the decreasing beta with solarcentric distance beyond the Martian orbit[60], more powerful jets than at Earth are expected in these environment. The effects of jets can be further amplified by the more compressible magnetospheres of the gas giants. How the jet effects seen at Earth play out at the outer planets remain to be seen in future studies. Moreover, as the outer planets host numerous satellites, one possible effect of jets is their

interaction with these satellites. For example, Titan's orbit is close to the dayside Kronian magnetopause and thus is prone to the impinging by jets. It was reported in cases the magnetopause was compressed by high dynamic pressure flows, Titan was located in the magnetosheath and experienced erosion of its remnant magnetic fields[61,62].

Jet influence on planetary magnetosphere aside, it is also beneficial to search for the jets in the flank part of the Jovian and Kronian magnetosheath. Although these jets may have minimal influence on the magnetopause, their examination, particularly in relation to their generation from high Mach number quasi-parallel shocks and their effects on particle accelerations, can contribute to our understanding of the broader issue of nonlinear shock dynamics.

Comparative studies of jets across planetary magnetosheath provide a new angle to shocks and magentospheres. Yet the available measurements are limited. While Saturn, Jupiter, Venus, Mercury have their own dedicated missions, due to limitations in instrumentation capabilities and orbital effects, the data cannot be used for statistical studies of magnetosheath jets. Mars is the only planet that accommodates large database. One other possibility is the downstream of interplanetary shocks throughout the heliosphere, which is planned in future research.

## Methods
### Data at jupiter
Voyager 2 data was used for Jovian jets. The magnetic field from Voyager 2 was measured using the magnetic field experiment (MAG)[63].

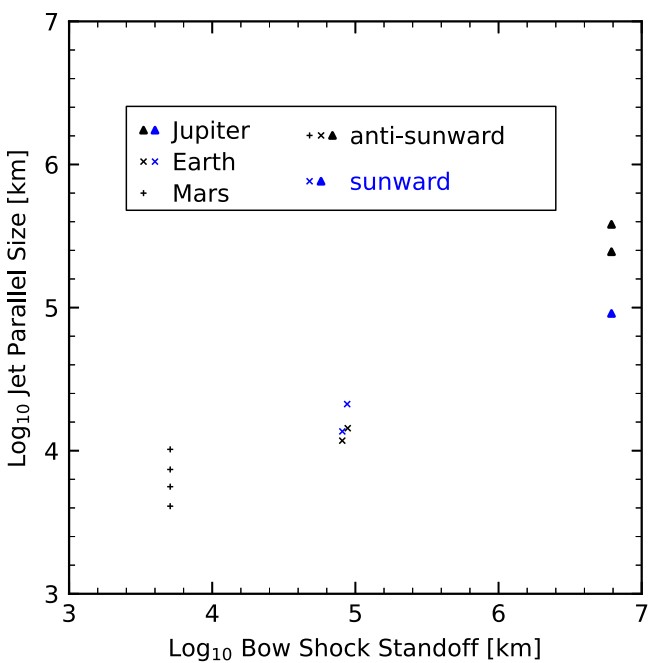

**Fig. 5 | Jet size parallel to its moving direction as a function of bow shock standoff distance for Mars, Earth, and Jupiter.** Blue (black) markers represent sunward (anti-sunward) jets. Source data are provided as a Source Data file.

The plasma data was measured using the plasma subsystem (PLS)[64]. The energetic ion (53–85 keV) data was provided by Low-Energy Charged Particle Investigation (LECP)[65]. Since Voyager 2 is a three-axis stabilized spacecraft and its plasma detector does not rotate, only a partial distribution function of ions can be measured. The two ion components were obtained by fitting the measurement using two independent Maxwellian distributions[27]. The dynamic pressure was calculated as $P_{dyn} = (n_1 + n_2)m_p v^2$. The trajectory of Voyager 2 in JSO coordinate system was obtained by using the positions of Voyager 2 and Jupiter in International Celestial Reference Frame (ICRF), which were retrieved from the Horizons system. The velocity used for each black arrow that represents a jet in Fig. 1a–c was found where the jet was at its highest speed.

### Standard deviation of magnetic field
The standard deviation of magnetic field in Fig. 1f is calculated following Karlsson et al.[35] by using 30-min window. The standard deviation $\sigma_i$ is calculated first for each magnetic field component $B_i$ ($i = 1, 2, 3$). The total standard deviation is then obtained by

$$\sigma(\mathbf{B}) = \left( \sum_i^3 \sigma_i^2 \right)^{1/2}. \tag{1}$$

### Data at mars and earth
MAVEN data was used for Martian jets. The magnetic field from MAVEN was measured by the MAVEN magnetic field investigation (MAG)[66]. The plasma data was measured using the Solar Wind Ion Analyzer (SWIA)[67]. THEMIS plasma data was used for the jets at Earth[68].

### Jets identification
At Jupiter, only Voyager 2 had passed the subsolar magnetosheath during its flyby and recorded the data as shown in Fig. 1. The jets duration was determined by visual selection of dynamic pressure pulse that is evident from a relatively calm background. Visual selection is also applied in determining the duration of jets at Mars and Earth.

### Voyager 2 trajectory in plasma and jet size
The trajectory of Voyager 2 in plasma was calculated by integrating the additive inverse of the ion velocity over time. The distances on top of Figs. 1 and 4 were calculated by integrating ion speed over time. The length of spacecraft trajectory during a jet observation was used as the parallel size of the jet. According to the schematic shown in Fig. 2, this would underestimate the parallel size. However, to compare jet sizes across planetary magnetosheath, it is optimal to use one unified method for size estimation. Alternatively, we may simply refer to this estimation as the size, rather than the parallel size, of jets. Since the plasma flow speeds are of similar magnitude across planetary magnetosheath, the parallel size here can also be interpreted as the duration of the jet.

### Magnetopause and bow shock positions
The positions of the bow shock and magnetopause at Jupiter shown in Fig. 1 are modeled using[28] and assuming a solar wind dynamic pressure of 0.19nPa. Since no simultaneous upstream observation was available at the time of downstream observation in the Jovian magnetosheath, the upstream parameters are chosen to roughly fit modeled positions with the location of bow shock and magnetopause crossings. Note that this fitting is only illustrative as the bow shock and magnetopause could have moved over a large range from the bow shock crossing to the magnetopause crossing. The modeled bow shock shown in Figs. 1 and 4 for Jupiter and Mars were used to obtain their standoff distance shown in Fig. 5. For observation at Earth, the standoff distance reported by OMNI at the time of encountering each jet was used[23].

## Data availability
Satellite mission data analyzed in this study are publicly available via the repositories of each satellite mission. Voyager 2, Cassini, MAVEN measurements are available at the PDS-PPI Node (https://pds-ppi.igpp.ucla.edu/). THEMIS measurements and OMNI data are available at CDAWeb (https://cdaweb.gsfc.nasa.gov/pub/data/). The plasma moments compiled by M. F. Thomsen et al.[69] from Cassini measurements was used in this study. This data set is available as supporting information to the publication at https://doi.org/10.1002/2018JA025214. The Voyager 2 and Jupiter positions in ICRF are available at the Horizons system (https://ssd.jpl.nasa.gov/horizons/app.html#/). The datasets generated in this study are provided in the Source Data file and also available from the corresponding author on request. Source data are provided with this paper.

## Code availability
The Python code used for this study consists of one routine, the Minimum Variance Analysis following[36]. It is available as Supplementary Software 1. It is also available at https://github.com/SpaceWalker162/space_database_analysis.

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

## Acknowledgements

We thank the Voyager team, Cassini team, MAVEN team, THEMIS team for providing data and support. We thank M. F. Thomsen et al. for providing the Cassini CAPS magnetosheath moments data set. We acknowledge the use of Horizons System, Coordinated Data Analysis Web (CDAWeb), the Planetary Plasma Interactions (PPI) Node of the Planetary Data System (PDS), and OMNI data. This work was supported by the National Natural Science Foundation of China, Grants No. 42130202 (C. S.), the National Key Research and Development Program of China, Grant No. 2022YFA1604600 (C. S.), and the National Natural Science Foundation of China under Grant No. 42330202 (S. W.).

## Author contributions

Y.Z. performed the data analysis and wrote the manuscript. S.R. and S.W. contributed to parts of the manuscript through reviews and edits. C.S. supervised the study and reviewed the manuscript. N.R. and L.M. contributed to the work by discussion. All authors contributed to the interpretation and discussion of the results.

## Competing interests

The authors declare no competing interests.
