## [Peer Review File · Nature Communications]

Magnetosheath jets at Jupiter and across the solar systemREVIEWER COMMENTS

Reviewer #1 (Remarks to the Author):

Review of "Jets in Planetary Magnetosheath" by Yufei Zhou, Chao Shen, Nian Ren, Lan Ma, Savvas Raptis, and Shan Wang

I think a revised version of this paper should be published in Nature Communications. It adds a new planet, Jupiter, to the list of solar system objects where jets have been detected, and that is a new result, which will have a significant impact on the research conducted within the growing magnetosheath jet community.

While the observations at Jupiter presented in this paper are convincing, I have concerns about the data from Saturn. The jet presented only comprises one single data point. The occasional fluke data point is often seen in moments of ion distributions, and even though it is possible that this corresponds to something real in the plasma, it could also be caused by a number of instrumental effects. To be convincing about a single point in moment data, one would have to analyse the underlying measurements on which the moments are based. The jet size estimate from a single point measurement is also fraught with uncertainty, and at best an upper limit can be specified. I cannot tell what the outcome of a more detailed analysis of the Saturnian data would be, but I think a paper about that would have to focus much more on technical details of the instrument.

So, for the present paper, I suggest you drop the Saturn part and present just Jupiter and the comparison to Mars and Earth. That is in itself is a major advance and enough to motivate publication in Nature Communications. And it will likely be a paper well received by the community and much cited in future.

Reviewer #2 (Remarks to the Author):

Magnetosheath jets are fast flow commonly observed in the magnetosheath, which modulate the solar wind-magnetosphere interaction. The authors conducted observations of anti-sunward and sunward magnetosheath jets in the Jovian and Kronian magnetosheath associated with discontinuities and compared their spatial scales across different planets. The observation results are interesting, but the interpretation has many issues. It is also not convincing why this work is significant enough for the general audience. Please see my detailed comments below.

About interpretation of the events:

In line 85, significant field and plasma variations are weak evidence to indicate quasi-parallel magnetosheath. How significant? Is there a quantified threshold? Even in the Earth's magnetosheath, it is not that straightforward to just use fluctuations to classify the magnetosheath (e.g., Karlsson et al., 2021). Additionally, around the first jet in Figure 1, the IMF is in the -X and -Y direction, so the local bow shock is quasi-perpendicular based on Figure 1a. After the first jet, the strong Bz component also tends to make the local bow shock in the equatorial plane quasi-perpendicular. I also see some anti-correlation between the magnetic field strength and plasma density, suggesting mirror mode fluctuations, which commonly exist in the quasi-perpendicular magnetosheath.

Karlsson, T., Raptis, S., Trollvik, H., & Nilsson, H. (2021). Classifying the magnetosheath behind the quasi-parallel and quasi-perpendicular bow shock by local measurements. *Journal of Geophysical Research: Space Physics*, 126, e2021JA029269. <https://doi.org/10.1029/2021JA029269>

In line 86, is θ cone angle? Why is it calculated from Bz (blue) rather than Bx (magenta) in Figure 1d? Or are you making wrong labels in Figure 1d? The magnetic field geometry here is very confusing.

In line 83, the statement about “reminiscent of the cores of HFAs” is not convincing. HFA remnants in the magnetosheath should be in a fast-mode sense. For example, in the three cited papers in line 84 and two cited papers in line 95 as well as many other observations papers like Hasegawa et al. (2012), Liu et al. (2020) and simulation papers like Wang et al. (2021), the HFA remnants in the magnetosheath all have a low field strength core. However, in Figure 1d, there is clear field strength increase in the “core”. Again, it looks like a mirror-mode structure. So, the observed jets could be due to the velocity and density fluctuations of mirror-mode structures (Blanco-Cano et al., 2020).

Hasegawa, H., Zhang, H., Lin, Y., Sonnerup, B. U. Ö., Schwartz, S. J., Lavraud, B., and Zong, Q.-G. (2012), Magnetic flux rope formation within a magnetosheath hot flow anomaly, *J. Geophys. Res.*, 117, A09214, doi:10.1029/2012JA017920.

Liu, T. Z., X. An, H. Zhang, and D. Turner (2020), Magnetospheric Multiscale (MMS) observations of foreshock transients at their very early stage, *ApJ*, 902:5 (15pp), <https://doi.org/10.3847/1538-4357/abb249>

Wang C-P, Wang X, Liu TZ, and Lin Y (2021). Impact of Foreshock Transients on the Flank Magnetopause and Magnetosphere and the Ionosphere. *Front. Astron. Space Sci.* 8:751244. doi: 10.3389/fspas.2021.751244

Blanco-Cano, X., Preisser, L., Kajdič, P., & Rojas-Castillo, D. (2020). Magnetosheath microstructure: Mirror mode waves and jets during southward IP magnetic field. *Journal of Geophysical Research: Space Physics*, 125, e2020JA027940. <https://doi.org/10.1029/2020JA027940>

In lines 92-93, even if it is an HFA, it is not correct to use MVAB or any methods across the entire HFA to calculate the normal. The two boundaries of HFAs have different orientations and not necessarily along the discontinuity normal (see previous simulation results). It is more like a triangle shape, not like that in Figure 2. Also, note that the MHD simulations by Otto and Zhang (2021) are not typical discontinuity-driven HFAs. They are driven by a density depletion region. Basically, the statement in line 95 is not supported.

In Figure 3, the jet only has one data point, so how is the duration calculated? How do you tell if the duration is actually much shorter than the time resolution of measurement? What if there are actually multiple jets within one data point? Thus, although Figure 5 looks reasonable, it is very questionable.

Again, θ in Figure 3e seems to be calculated from B_z (blue) in Figure 3d. Also, is this quasi-parallel or quasi-perpendicular magnetosheath?

About the importance of this topic:

In line 32, the authors must show evidence why “the formation of jets is an important and integral part of these dynamics” mentioned in Turner et al. (2018) and Russell et al. (1982). It is not clear to me, let alone to the general audience.

In lines 140-141, this statement is misleading. Jets do not “only” form in the quasi-parallel region. They just have a much higher occurrence rate in the quasi-parallel region. Also, “rippling” is not the only mechanism. Please check the cited review paper for details. Also, Kajdič et al. (2021) explained some mechanisms for jets in the quasi-perpendicular region.

Statement in lines 142-144 is not well supported. 1. Parker spiral is just an average sense of IMF configuration. Due to the high variation of the IMF, the subsolar quasi-parallel region can still occur at Jupiter’s and Saturn’s bow shock. It just occurs less frequently than at Earth’s bow shock. (Also note that the ion foreshock is not the same as quasi-parallel region as it can occur in the quasi-perpendicular region). 2. Why is only the subsolar region considered? Why is the broad flank region not important? 3. As already mentioned, the quasi-perpendicular region can still have magnetosheath jets.

In lines 149-151, in Turner et al. (2018), the acceleration is due to Fermi within an HFA. It could be interesting to see whether the remnants of HFAs in the magnetosheath could play a role in the acceleration process, but I do not see how identifying jets from the HFA remnants contributes.

There has been no study about this for Earth's environment yet. Additionally, due to what are mentioned in lines 146-147, even though there are discontinuity-bow shock interactions, HFAs as well as associated particle acceleration less likely occur in the subsolar sections. Especially, there is no sufficient evidence of HFAs in this study. Thus, although I agree studying particle acceleration at high Mach number shocks is important, I do not see how this study contributes.

First of all, we would like to extend our gratitude to you on your inspiring and detailed comments and recommendations.

The lining number mentioned in this Response To Reviewers refers to the updated manuscript without tracking.

I. RESPONSE TO REVIEWER 1

Reviewer #1 (Remarks to the Author):

Review of “Jets in Planetary Magnetosheath” by Yufei Zhou, Chao Shen, Nian Ren, Lan Ma, Savvas Raptis, and Shan Wang

I think a revised version of this paper should be published in Nature Communications. It adds a new planet, Jupiter, to the list of solar system objects where jets have been detected, and that is a new result, which will have a significant impact on the research conducted within the growing magnetosheath jet community.

While the observations at Jupiter presented in this paper are convincing, I have concerns about the data from Saturn. The jet presented only comprises one single data point. The occasional fluke data point is often seen in moments of ion distributions, and even though it is possible that this corresponds to something real in the plasma, it could also be caused by a number of instrumental effects. To be convincing about a single point in moment data, one would have to analyse the underlying measurements on which the moments are based. The jet size estimate from a single point measurement is also fraught with uncertainty, and at best an upper limit can be specified. I cannot tell what the outcome of a more detailed analysis of the Saturnian data would be, but I think a paper about that would have to focus much more on technical details of the instrument.

So, for the present paper, I suggest you drop the Saturn part and present just Jupiter and the comparison to Mars and Earth. That is in itself is a major advance and enough to motivate publication in Nature Communications. And it will likely be a paper well received by the community and much cited in future.

Reply: We thank the reviewer for appreciation, and we agree with the reviewer on the Saturnian event. A data point is less convincing. But even without plasma data, the variation of magnetic field also suggested that a special structure was there. However, the low resolution of plasma data indeed makes the event less valuable. Thus we have decided to move the Saturnian observation to Supplementary Information as a preliminary result.

II. RESPONSE TO REVIEWER 2

Reviewer #2 (Remarks to the Author):

Magnetosheath jets are fast flow commonly observed in the magnetosheath, which modulate the solar wind-magnetosphere interaction. The authors conducted observations of anti-sunward and sunward magnetosheath jets in the Jovian and Kronian magnetosheath associated with discontinuities and compared their spatial scales across different planets. The observation results are interesting, but the interpretation has many issues. It is also not convincing why this work is significant enough for the general audience. Please see my detailed comments below.

A. About interpretation of the events

In line 85, significant field and plasma variations are weak evidence to indicate quasi-parallel magnetosheath. How significant? Is there a quantified threshold? Even in the Earth’s magnetosheath, it is not that straightforward to just use fluctuations to classify the magnetosheath (e.g., Karlsson et al., 2021). Additionally, around the first jet in Figure 1, the IMF is in the -X and -Y direction, so the local bow shock is quasi-perpendicular based on Figure 1a. After the first jet, the strong Bz component also tends to make the local bow shock in the equatorial plane quasi-perpendicular.

Reply: We agree with the reviewer, and indeed a quantitative method as developed by *Karlsson et al. (2021)*, to identify quasi-parallel/quasi-perpendicular magnetosheath, is more appropriate than visual inspection. To accommodate this classification, high energy ion flux and standard deviation of magnetic field have been added to figure 1. The region after the second and third jets shows a much lower level of magnetic standard deviation and high energy ion flux, in contrast to the region before the jets. The work by Karlsson et al. suggests that high energy ion flux is a good indicator for classifications of both Qpar-Qperp geometries and foreshock-noforeshock conditions, while magnetic variance is a good indicator for classifying Qpar-Qperp geometries alone. Thus, based on both indicators, it appears that the plasma before the two-jets originates from a quasi-parallel shock crossing while the plasma after the jet from a quasi-perpendicular one. Discussion has been added to lines 95–101

I also see some anti-correlation between the magnetic field strength and plasma density, suggesting mirror mode fluctuations, which commonly exist in the quasi-perpendicular magnetosheath.

Reply: We thank the reviewer for pointing out this possibility. Indeed, mirror mode structures are common in the Jovian and Kronian magnetosheath. As for the Jovian case, previous statistics show that for 62% of the time of observation the magnetosheath shows magnetic signatures of mirror mode. However, these structures are 20-120 seconds (*Erdős and Balogh, 1996; Joy et al., 2006*) and 20 proton gyroradii ($\sim 0.15R_J$) (*Erdős and Balogh, 1996; Hasegawa and Tsurutani, 2011*) respectively, thus not likely being responsible for the jets reported here. We have added a new figure (Fig. 3 in the updated manuscript) to show the zoomed in view of the downstream HFA. In this figure the positive correlation of magnetic field and density at the scale of the jets in the third jet is visible. This suggests a fast mode nature, which is consistent with the interpretation of being the boundary of a HFA. The correlation in the second jet is less clear. Discussion about mirror modes has been added to lines 113–121 in the manuscript without tracking.

In line 86, is cone angle? Why is it calculated from B_z (blue) rather than B_x (magenta) in Figure 1d? Or are you making wrong labels in Figure 1d? The magnetic field geometry here is very confusing.

Reply: As indicated in the caption, θ and ϕ are the second and third spherical coordinates of the magnetic field vector. They are defined through

$$\begin{aligned} B_x &= B \sin \theta \cos \phi \\ B_y &= B \sin \theta \sin \phi \\ B_z &= B \cos \theta \end{aligned} \tag{1}$$

In line 83, the statement about “reminiscent of the cores of HFAs” is not convincing. HFA remnants in the magnetosheath should be in a fast-mode sense. For example, in the three cited papers in line 84 and two cited papers in line 95 as well as many other observations papers like *Hasegawa et al. (2012)*, *Liu et al. (2020)* and simulation papers like *Wang et al. (2021)*, the HFA remnants in the magnetosheath all have a low field strength core. However, in Figure 1d, there is clear field strength increase in the “core”.

Reply: We have added a new figure, Fig. 3 in the updated manuscript, showing the details of the second and third jets.

While we agree with the reviewer that HFAs observed upstream of the shock typically have a lower magnetic field at their core, sub-structures exist in these HFAs, which complicate the structures. For example, $|B|$ may exhibit an increase in the center of upstream HFAs (see, e.g. the THC observation shown by green line in the fourth panel of Fig. 3 *Archer et al., 2014*). HFAs further evolve and deform in the magnetosheath, which may form peaks in the center (e.g., Figure 2 in *Safrankova et al., 2012*; Panel d of Figure 3 of *Hasegawa et al. (2012)*), or multiple peaks and dips (Figure 4 in *Eastwood et al. (2008)*). In our event, a short $|B|$ decrease exists, which is better seen in Figure 3 now. Together with other plasma conditions (low P_{dyn} , low density, high temperature, velocity deflection), the structure is identified as an HFA remnant in the magnetosheath.

Again, it looks like a mirror-mode structure. So, the observed jets could be due to the velocity and density fluctuations of mirror-mode structures (*Blanco-Cano et al., 2020*).

Reply: The issue about mirror mode is addressed on lines 113–121

In lines 92-93, even if it is an HFA, it is not correct to use MVAB or any methods across the entire HFA to calculate the normal. The two boundaries of HFAs have different orientations and not necessarily along the discontinuity normal (see previous simulation results). It is more like a triangle shape, not like that in Figure 2.

Reply: Thank you for pointing out this. We apologize for not having made this point clear. Since in many studies the two boundaries of HFAs have different orientations, we have excluded the boundaries (i.e. the jets) in the MVAB calculation. The region denoted with HFA in the original manuscript should be the core of HFA. The resultant direction from MVAB is the minimum variance direction of the HFA core. Modification has been made to clarify this point (see lines 104–106).

Also, note that the MHD simulations by *Otto and Zhang (2021)* are not typical discontinuity-driven HFAs. They are driven by a density depletion region. Basically, the statement in line 95 is not supported.

Reply: We have removed the reference to *Otto and Zhang (2021)*, and added an extra reference to the Fig. 2 of the other paper by *Lin (2002)* (see line 114). *Lin's* simulation used tangential discontinuity, which is regarded as the most common type of discontinuities to interact with bow shock and to form HFA. Here we attach a figure (Fig.1) adapted from the Figure 2 of the paper. The third panel shows the temperature and the HFA can be identified from it. The second panel shows flow velocity, with arrow representing both direction and magnitude. The HFA moves from left top to right bottom. If a spacecraft is located in the magnetosheath and swept through by the HFA in her simulation, its trajectory in the (approximate) rest frame of the HFA core would look like the blue-red-blue arrow shown in the second panel, which is similar to the trajectory reconstructed for our Jovian observation shown in Fig.2 in our manuscript. And the velocity observed by the spacecraft in the simulation would look like the Jovian data shown in Fig.1 (h) in the updated manuscript. *Lin* plotted the data to be observed by a virtual spacecraft in the magnetosheath in the second row of Fig.6 of her paper (Fig. 2 here). Indeed, the velocities shown in the panel on the

FIG. 1. Fig.2 from Lin (2002) with our illustration. Contours of B_y , ion flow vectors plotted against the contours of ion flow speed in the xz plane, and the ion temperature contours show the structure of HFA. Blue and red lines with a arrow head shows the trajectory of a spacecraft in the HFA if it is to be swept through by the HFA.

second row and second column in Fig. 2 are similar to our Jovian observation of the magnetosheath HFA. Density and temperature observations are also similar. The magnetic field is not the same, but both the simulation and observation show an increased magnetic field in the core (third column), and a less variant B_z component than other magnetic vector components (first column), which point to the same minimum variance direction of the structures. In summary, the Jovian observation does show similarities to the simulation by Lin.

In Figure 3, the jet only has one data point, so how is the duration calculated? How do you tell if the duration is

FIG. 2. Fig.6 from *Lin* (2002). Time variations of B_y (heavy lines), B_z (light), V_x (heavy), V_z (light), N (heavy), B (light), T_{\parallel} (heavy), and T_{\perp} (light) at three locations along $\theta = 80^\circ$ at $r = 22R_E$ in the upstream solar wind, $r = 14R_E$ in the magnetosheath, and $r = 8R_E$ near the magnetopause.

actually much shorter than the time resolution of measurement? What if there are actually multiple jets within one data point? Thus, although Figure 5 looks reasonable, it is very questionable.

Again, in Figure 3e seems to be calculated from B_z (blue) in Figure 3d. Also, is this quasi-parallel or quasi-perpendicular magnetosheath?

Reply: The sampling time of the data point is used for the duration now. Although this may not be of high accuracy, it suffices to provide an estimation of the duration to be compared with jet duration at other planets, since their orders of magnitude are quite different. While we agree that such an estimate represents an upper limit of the jet size, we would like to mention a weak evidence that the magnetic field observation (panel e of Supplementary Fig. 1) suggests that the jet is indeed of this size. Due to the resolution concern raised by the reviewer, and as suggested by the other reviewer, we have move this part, along with caveats, to the Supplementary Information as a preliminary result. θ and ϕ are the spherical coordinates of the magnetic field vector. Our conclusion is that the region is of quasi-parallel origin, according to the geometry plotted in panel (a).

B. About the importance of this topic

In line 32, the authors must show evidence why “the formation of jets is an important and integral part of these dynamics” mentioned in Turner et al. (2018) and Russell et al. (1982). It is not clear to me, let alone to the general audience.

Reply: We agree with the reviewer and decided to remove this part from the introduction. More detail regarding the relations among jet, HFA, acceleration has been added to the Discussion section (see lines 159–169).

In lines 140-141, this statement is misleading. Jets do not “only” form in the quasi-parallel region. They just have a much higher occurrence rate in the quasi-parallel region. Also, “rippling” is not the only mechanism. Please check the cited review paper for details. Also, Kajdič et al. (2021) explained some mechanisms for jets in the quasi-perpendicular region.

Reply: We thank the reviewer for pointing out this. Our original intention is to stress “early studies” and the “prevailing” but not necessarily full picture. It is true, that the community usually believes that magnetosheath jets require an upstream foreshock and sequentially quasi-parallel shock to be present. However, as shown by recent statistical studies (*Goncharov et al.*, 2020; *Raptis et al.*, 2020; *Vuorinen et al.*, 2019, 2023) there is a non-trivial portion of jets observed in the quasi-perpendicular magnetosheath. The mechanisms responsible for their presence may include magnetic reconnection, flux tubes, mirror mode waves (*Kajdič et al.*, 2021) and shock-discontinuity interactions (*Zhou et al.*, 2023). In this work, we mainly discussed generation mechanisms associated to shock dynamics, the referencing to other works was indeed limited. We have now modified the sentence to acknowledge jets originated from non-shock processes and structures in quasi-perpendicular magnetosheath. Please see line 152-154

Statement in lines 142-144 is not well supported. 1. Parker spiral is just an average sense of IMF configuration. Due to the high variation of the IMF, the subsolar quasi-parallel region can still occur at Jupiter’s and Saturn’s bow shock. It just occurs less frequently than at Earth’s bow shock. (Also note that the ion foreshock is not the same as quasi-parallel region as it can occur in the quasi-perpendicular region). 2. Why is only the subsolar region considered?

Why is the broad flank region not important? 3. As already mentioned, the quasi-perpendicular region can still have magnetosheath jets.

Reply:

1. We have modified the text. Please check lines 154–158. Regarding ion foreshock, those structures responsible for shock reformation and shock curvatures are usually present in the inner part of ion foreshock. Thus although the boundary of ion foreshock can extend to the quasi-perpendicular part of bow shocks, the structures leading to jet generation are not typically connected to these shock geometries. Hence we believe that the classification of jets based on quasi-perpendicular and quasi-parallel geometries can still be used without violating the statistically expected picture. It should be noted that while the terms "quasi-parallel" and "quasi-perpendicular" are used throughout the text, we do not follow necessarily the rigid theoretical threshold of 45 degrees (sec. 5.1 of *Balogh and Treumann*, 2013). Here we use the quasi-parallel region as the one which the presence of a significant ion foreshock dictates the dynamics of the downstream region (presence of ULF waves, turbulence, non-linearly evolved waves etc.). On the other hand, when pointing to quasi-perpendicular we refer to the environments in which the reflected particles are contained within the shock's foot region, making the transitions smoother and the downstream regions less turbulent.).
2. We agree that flank jets are also important and that indeed they appear to form as shown in Earth's magnetosheath. We have added discussion on the potential significance of future studies on flank jets in the Jovian and magnetosheath (please see 185–188). Since this is the first report of jets in Jovian magnetosheath, we focus on subsolar region to follow the presently available literature. Most jet studies to date concentrate on the influence of jets on geomagnetism. Since the jets formed from the flank part of the terrestrial bow shock most likely have limited impact with the magnetopause, they are less explored. As a natural extension toward planetary systems, we also stress the subsolar region. In the future, we could also extend the studies to flank regions.
3. We have now acknowledged at the beginning of this paragraph the jet formations that are related to non-shock processes (Please see lines 154–158). Regarding mirror mode structures in the Jovian magnetosheath, while we mention them in the manuscript, the scales appear to be smaller than these of the jet observations.

In lines 149-151, in Turner et al. (2018), the acceleration is due to Fermi within an HFA. It could be interesting to see whether the remnants of HFAs in the magnetosheath could play a role in the acceleration process, but I do not see how identifying jets from the HFA remnants contributes. There has been no study about this for Earth's environment yet.

Reply: We agree with the reviewer, and decided to remove the associated lines. Indeed, in the event reported here, enhancement of energetic particles is shown around the remnant of the HFA. We have mentioned this enhancement now on lines 166–169. But its further investigation is out of the scope of the present study.

Additionally, due to what are mentioned in lines 146-147, even though there are discontinuity-bow shock interactions, HFAs as well as associated particle acceleration less likely occur in the subsolar sections.

Reply: Simulation and statistics suggest that shock–discontinuity interactions that produce HFAs can occur at both quasi-perpendicular and quasi-parallel shocks (*Lin*, 2002; *Thomas et al.*, 1991; *Wang et al.*, 2013). Thus although the subsolar sections are frequently perpendicular, it does not undermine the conditions of shock–discontinuity interactions..

Especially, there is no sufficient evidence of HFAs in this study. Thus, although I agree studying particle acceleration at high Mach number shocks is important, I do not see how this study contributes.

Reply: As mentioned above, we have now provided magnetic field standard deviation and energetic ion flux to support the identification of the transition from quasi-parallel to quasi-perpendicular magnetosheath. Such transitions are typically associated with discontinuities for which their interaction with a bow shock can lead to a HFA. Other evidence includes the heated, density-depleted, and flow-deflected core region, the fast mode evident in the trailing boundary, and the density pileup in the two boundaries. While we agree, that we cannot point to the formation of an HFA since this study focuses on the downstream signals connected to jet observations, there is good support interpreting the observations as being associated to the presence of a downstream HFA.

We acknowledge that this study does not contribute directly to the study of particle acceleration at high Mach number shocks, as this is out of the scope of this study. Instead, since we have presented a new picture of jet formation from shock–discontinuity interaction, we point to a potential connection between the study of particle acceleration and the study of jets (see lines 163–169). These adjacent fields combined could help us understand more in the future. Our work effectively shows the connection of these phenomena (HFA, shock interaction, and jet formation). Most importantly we highlight their universal presence throughout the heliosphere by showing their existence in other planetary environments. We agree on the last part of the reviewer's comment and we clarified in the discussion (see

lines 163–169 and 185–188) that while this study does not directly address particle acceleration, it paves the way for future research to investigate these aspects both at Earth and in other planetary systems.

-
- Archer, M. O., D. L. Turner, J. P. Eastwood, T. S. Horbury, and S. J. Schwartz, The role of pressure gradients in driving sunward magnetosheath flows and magnetopause motion, *J. Geophys. Res.: Space Phys.*, *119*(10), 8117–8125, doi:10.1002/2014JA020342, 2014.
- Balogh, A., and R. A. Treumann, *Physics of Collisionless Shocks: Space Plasma Shock Waves*, ISSI Scientific Report Series 12, 1 ed., Springer-Verlag New York, 2013.
- Eastwood, J. P., et al., THEMIS observations of a hot flow anomaly: Solar wind, magnetosheath, and ground-based measurements, *Geophysical Research Letters*, *35*(17), L17S03, doi:10.1029/2008GL033475, 2008.
- Erdős, G., and A. Balogh, Statistical properties of mirror mode structures observed by ulysses in the magnetosheath of jupiter, *Journal of Geophysical Research: Space Physics*, *101*(A1), 1–12, doi:10.1029/95JA02207, 1996.
- Goncharov, O., H. Gunell, M. Hamrin, and S. Chong, Evolution of high-speed jets and plasmoids downstream of the quasi-perpendicular bow shock, *J. Geophys. Res.: Space Phys.*, *125*(6), e2019JA027,667, doi:https://doi.org/10.1029/2019JA027667, e2019JA027667 2019JA027667, 2020.
- Hasegawa, A., and B. T. Tsurutani, Mirror mode expansion in planetary magnetosheaths: Bohm-like diffusion, *Physical Review Letters*, *107*(24), 245,005, doi:10.1103/PhysRevLett.107.245005, 2011.
- Hasegawa, H., H. Zhang, Y. Lin, B. U. . Sonnerup, S. J. Schwartz, B. Lavraud, and Q.-G. Zong, Magnetic flux rope formation within a magnetosheath hot flow anomaly, *Journal of Geophysical Research: Space Physics*, *117*(A9), n/a–n/a, doi: 10.1029/2012JA017920, 2012.
- Joy, S. P., M. G. Kivelson, R. J. Walker, K. K. Khurana, C. T. Russell, and W. R. Paterson, Mirror mode structures in the jovian magnetosheath, *Journal of Geophysical Research*, *111*(A12), doi:10.1029/2006JA011985, 2006.
- Kajdič, P., S. Raptis, X. Blanco-Cano, and T. Karlsson, Causes of jets in the quasi-perpendicular magnetosheath, *Geophys. Res. Lett.*, *48*(13), e2021GL093,173, doi:10.1029/2021GL093173, 2021.
- Karlsson, T., S. Raptis, H. Trollvik, and H. Nilsson, Classifying the magnetosheath behind the quasi-parallel and quasi-perpendicular bow shock by local measurements, *J. Geophys. Res.: Space Phys.*, *126*(9), e2021JA029,269, doi: https://doi.org/10.1029/2021JA029269, e2021JA029269 2021JA029269, 2021.
- Lin, Y., Global hybrid simulation of hot flow anomalies near the bow shock and in the magnetosheath, *Planetary and Space Science*, *50*(5-6), 577–591, doi:10.1016/S0032-0633(02)00037-5, 2002.
- Otto, A., and H. Zhang, Bow shock transients caused by solar wind dynamic pressure depletions, *Journal of Atmospheric and Solar-Terrestrial Physics*, *218*, 105,615, doi:10.1016/j.jastp.2021.105615, 2021.
- Raptis, S., T. Karlsson, F. Plaschke, A. Kullen, and P.-A. Lindqvist, Classifying magnetosheath jets using MMS: Statistical properties, *Journal of Geophysical Research: Space Physics*, *125*(11), e2019JA027,754, doi: https://doi.org/10.1029/2019JA027754, 2020.
- Thomas, V. A., D. Winske, M. F. Thomsen, and T. G. Onsager, Hybrid simulation of the formation of a hot flow anomaly, *J. Geophys. Res.: Space Phys.*, *96*(A7), 11,625–11,632, doi:https://doi.org/10.1029/91JA01092, 1991.
- Vuorinen, L., H. Hietala, and F. Plaschke, Jets in the magnetosheath: IMF control of where they occur, *Ann. Geophys.*, *37*(4), 689–697, doi:10.5194/angeo-37-689-2019, 2019.
- Vuorinen, L., H. Hietala, A. T. LaMoury, and F. Plaschke, Solar wind parameters influencing magnetosheath jet formation: Low and high IMF cone angle regimes, *Journal of Geophysical Research: Space Physics*, *128*(10), doi:10.1029/2023JA031494, 2023.
- Wang, S., Q. Zong, and H. Zhang, Cluster observations of hot flow anomalies with large flow deflections: 2. bow shock geometry at HFA edges, *Journal of Geophysical Research: Space Physics*, *118*(1), 418–433, doi:10.1029/2012JA018204, 2013.
- Zhou, Y., C. Shen, and Y. Ji, Undulated shock surface formed after a shock-discontinuity interaction, *Geophys. Res. Lett.*, *50*(10), e2023GL103,848, doi:https://doi.org/10.1029/2023GL103848, e2023GL103848 2023GL103848, 2023.

REVIEWER COMMENTS

Reviewer #1 (Remarks to the Author):

Review of the revised version of "Jets in Planetary Magnetosheath" by Yufei Zhou, Savvas Raptis, Shan Wang, Chao Shen, Nian Ren, and Lan Ma

I think you can publish this now. My concern was the single data point in the observation at Saturn. The authors have dealt with that by moving the part about Saturn into an appendix and calling it preliminary results. It is now clear how uncertain that data point is. I have no objection to publishing the manuscript as is, but I think it needs to be an editorial decision whether to keep that appendix or to cut it.

In any case, I think the word "Jupiter" should appear in the title of the paper, as the discovery of jets at Jupiter is the main result.

Reviewer #2 (Remarks to the Author):

Thank you for addressing my comments. After some further clarification, the paper may be suitable for publication.

I agree that there can be waves, substructures, etc. that can enhance field strength in HFA cores, but the overall trend of field strength decrease should be at least seen (especially after smoothing B vector), rather than very localized, transient decrease with an overall trend of increase shown here. In Lin (2002), the field strength changes, correlated with the density changes, show clear trend of decrease in the core as well as increase at two boundaries, which is quite different from this study. Weak B component variation simply relies on the magnetic field and discontinuity geometries, which can be in a wide range of directions and is not a universal property. Considering that the observations of HFAs at Jovian bow shock are still very limited and we do not have sufficient understanding, maybe it is indeed an HFA remnant, and this field strength change is due to some special mechanisms at Jupiter. However, because it is the first time to claim HFAs in the Jovian magnetosheath, stricter identifications must be performed and there should not be any ambiguity such as the field strength variation. I understand that this paper focuses on magnetosheath jets rather than HFAs. So, I would suggest the authors say HFAs as a promising candidate rather than a definite conclusion to avoid those debates. For example, change "was a downstream HFA" to something like "could be a downstream HFA" or "ensembled a downstream HFA".

The reason I asked to clarify the connection with Turner et al. (2018) is to enhance the significance of this study for general readers. Why would readers other than the space community want to care about magnetosheath jets? Deleting it does not help. Here I provide an example for you:

Shock acceleration is a fundamental source of energetic particles throughout the universe, yet its understanding is still insufficient. Magnetosheath jets have been shown to accelerate particles. This means that transient structures both upstream and downstream of shocks that have been neglected in shock acceleration theories can result in additional acceleration, providing seed populations and enhancing shock acceleration efficiency (as discussed in Liu et al. 2020 <https://agupubs.onlinelibrary.wiley.com/doi/full/10.1029/2019JA027710>). This is especially true for discontinuity-driven jets because discontinuities have been shown to drive both foreshock transients and magnetosheath jets. Observing them across different planets can prove that this scenario could work in general shock environments such as high Mach number astrophysical shocks.

MVAB across the core is not a common way to estimate the orientation. (1) Its direction is spatial dependent because around the part closer to a compressional boundary, the local MVAB direction is more aligned with that boundary normal. (2) Waves, fluctuations, and substructures inside HFA

cores can strongly contaminate the MVAB direction. To double check the reliability of this direction: because HFAs are roughly along the discontinuity surface, assuming the discontinuity is a tangential discontinuity (typical for HFAs), is the cross product of magnetic field before and after the event consistent with the MVAB normal? If you use MVAB at leading and trailing edges of the event, are the normals reasonable?

If you want to change the definition of "quasi-parallel" and "quasi-perpendicular" in the text, it should be clearly stated, otherwise terms such as "foreshock" and "non-foreshock" can be used.

In the caption of Figure 1: 53-85 eV -> 52-85 keV, right?

Thank you again for your suggestions and comments, which we find indeed helpful for improving the manuscript. The lining number mentioned in this Response To Reviewers refers to the updated manuscript **with** tracking.

I. RESPONSE TO REVIEWER 1

Reviewer #1 (Remarks to the Author):

Review of the revised version of “Jets in Planetary Magnetosheath” by Yufei Zhou, Savvas Raptis, Shan Wang, Chao Shen, Nian Ren, and Lan Ma

I think you can publish this now. My concern was the single data point in the observation at Saturn. The authors have dealt with that by moving the part about Saturn into an appendix and calling it preliminary results. It is now clear how uncertain that data point is. I have no objection to publishing the manuscript as is, but I think it needs to be an editorial decision whether to keep that appendix or to cut it.

In any case, I think the word “Jupiter” should appear in the title of the paper, as the discovery of jets at Jupiter is the main result.

Reply: We agree with the reviewer and decided to rename the study “Magnetosheath jets at Jupiter and Across the Solar System”

II. RESPONSE TO REVIEWER 2

Reviewer #2 (Remarks to the Author):

Thank you for addressing my comments. After some further clarification, the paper may be suitable for publication.

I agree that there can be waves, substructures, etc. that can enhance field strength in HFA cores, but the overall trend of field strength decrease should be at least seen (especially after smoothing B vector), rather than very localized, transient decrease with an overall trend of increase shown here. In Lin (2002), the field strength changes, correlated with the density changes, show clear trend of decrease in the core as well as increase at two boundaries, which is quite different from this study. Weak B component variation simply relies on the magnetic field and discontinuity geometries, which can be in a wide range of directions and is not a universal property. Considering that the observations of HFAs at Jovian bow shock are still very limited and we do not have sufficient understanding, maybe it is indeed an HFA remnant, and this field strength change is due to some special mechanisms at Jupiter. However, because it is the first time to claim HFAs in the Jovian magnetosheath, stricter identifications must be performed and there should not be any ambiguity such as the field strength variation. I understand that this paper focuses on magnetosheath jets rather than HFAs. So, I would suggest the authors say HFAs as a promising candidate rather than a definite conclusion to avoid those debates. For example, change “was a downstream HFA” to something like “could be a downstream HFA” or “ensembled a downstream HFA”.

Reply: We agree with the reviewer and have made modifications accordingly throughout the text (see lines 56 and 113). HFAs at Jupiter may be different from those at Earth. As shown in an upstream HFA identified previously using Juno data *Valek et al.* (2017) (Fig. 1 here), the magnetic field had an overall increase in the core, except a localized decrease suggesting the existence of a current sheet. The event reported in this study showed similar magnetic signature. We have now mentioned this magnetic signature in lines 101-102

The reason I asked to clarify the connection with Turner et al. (2018) is to enhance the significance of this study for general readers. Why would readers other than the space community want to care about magnetosheath jets? Deleting it does not help. Here I provide an example for you: Shock acceleration is a fundamental source of energetic particles throughout the universe, yet its understanding is still insufficient. Magnetosheath jets have been shown to accelerate particles. This means that transient structures both upstream and downstream of shocks that have been neglected in shock acceleration theories can result in additional acceleration, providing seed populations and enhancing shock acceleration efficiency (as discussed in Liu et al. 2020 <https://agupubs.onlinelibrary.wiley.com/doi/full/10.1029/2019JA027710>). This is especially true for discontinuity-driven jets because discontinuities have been shown to drive both foreshock transients and magnetosheath jets. Observing them across different planets can prove that this scenario could work in general shock environments such as high Mach number astrophysical shocks.

Reply: We thank the reviewer for providing this great example. A paragraph modified from this example has been added to lines 47-54.

MVAB across the core is not a common way to estimate the orientation. (1) Its direction is spatial dependent because around the part closer to a compressional boundary, the local MVAB direction is more aligned with that boundary normal. (2) Waves, fluctuations, and substructures inside HFA cores can strongly contaminate the MVAB direction. To double check the reliability of this direction: because HFAs are roughly along the discontinuity surface,

FIG. 1. Fig. 2 from Valek et al. (2017).

assuming the discontinuity is a tangential discontinuity (typical for HFAs), is the cross product of magnetic field before and after the event consistent with the MVAB normal? If you use MVAB at leading and trailing edges of the event, are the normals reasonable?

Reply: The MVAB produced a direction of $[-0.171, 0.040, 0.984]$. Fig. 2 shows two possible sets of pre-discontinuity and post-discontinuity regions with magenta and green shadings respectively. The two sets result in the magnetic field conditions and discontinuity parameters shown in Tab. I. The first set (magenta) was so chosen outside of the entire event and not to reach the rotations of magnetic field around 03:30 (hr:min) and 06:00, which are visible in the spherical coordinates of magnetic field vector shown in Fig. 2 (b). This set results in a large magnetic shear angle of 73.1° . The second set (green) was chosen to be near the event, which gave a relatively small shear angle of 33.0° . The normal directions of the discontinuity calculated from both sets are close to that obtained from MVAB method ($16^\circ \sim 17^\circ$). These results have been added to Supplementary Information.

MVAB cannot be applied to the jets here, as there was essentially no minimum variance direction (λ ratio:1.71 and 1.15). It is in fact visible from Fig. 2 (a) that all magnetic components in the jets have similar variances. However, in the core B_z is less variant than the other two components, resulting in a minimum variance direction with a dominant z -component.

If you want to change the definition of “quasi-parallel” and “quasi-perpendicular” in the text, it should be clearly stated, otherwise terms such as “foreshock” and “non-foreshock” can be used.

Reply: We thank the reviewer for this suggestion. After serious consideration, we think we should have no intention to change the definition of a standard notion, which would otherwise lead to confusion. The current use of “quasi-parallel” and “quasi-perpendicular” throughout the text is consistent with the definition based on 45° and consistent

FIG. 2. Magenta color shading delineates the pre-event and post-event data used for calculating TD normal.

with their use by previous studies on jets, which usually use "quasi-parallel" and "quasi-perpendicular" rather than "foreshock" and "non-foreshock" *Goncharov et al. (2020)*; *Plaschke et al. (2018)*; *Raptis et al. (2020)*; *Vuorinen et al. (2023)* to classify jets and bow shocks. Also, both high energy ion flux and magnetic field variance, utilized to identify regions before and after the structure in this study, are good indicators to distinguish between quasi-parallel and quasi-perpendicular geometry *Karlsson et al. (2021)*. However, we have added the connection between foreshock dominance and the different shock orientation in lines 31-34, 168-170.

In the caption of Figure 1: 53-85 eV \rightarrow 52-85 keV, right?

TABLE I. magnetic field conditions and discontinuity parameters.

Parameter	Value
\mathbf{n}_{MVAB}	[-0.171, 0.040, 0.984]
Set 1 (magenta color shading in Fig. 2)	
\mathbf{B}_{pre}	[1.09, -0.994, 0.670] nT
\mathbf{B}_{post}	[-0.819, 1.56, 0.270] nT
$\theta_{\text{BpreBpost}}$	73.1°
$\mathbf{n}_{\text{BcrossB}}$	[-0.281, 0.305, 0.910]
Normal angle difference from \mathbf{n}_{MVAB}	17.0°
Set 2 (green color shading in Fig. 2)	
\mathbf{B}_{pre}	[-1.78, -0.232, -0.562] nT
\mathbf{B}_{post}	[-0.960, -0.870, -0.481] nT
$\theta_{\text{BpreBpost}}$	33.0°
$\mathbf{n}_{\text{BcrossB}}$	[-0.267, -0.224, 0.937]
Normal angle difference from \mathbf{n}_{MVAB}	16.4°

Reply: Thank you for pointing out this. Correction has been made.

Goncharov, O., H. Gunell, M. Hamrin, and S. Chong, Evolution of high-speed jets and plasmoids downstream of the quasi-perpendicular bow shock, *J. Geophys. Res.: Space Phys.*, 125(6), e2019JA027,667, doi:<https://doi.org/10.1029/2019JA027667>, e2019JA027667 2019JA027667, 2020.

Karlsson, T., S. Raptis, H. Trollvik, and H. Nilsson, Classifying the magnetosheath behind the quasi-parallel and quasi-perpendicular bow shock by local measurements, *J. Geophys. Res.: Space Phys.*, 126(9), e2021JA029,269, doi:<https://doi.org/10.1029/2021JA029269>, e2021JA029269 2021JA029269, 2021.

Plaschke, F., et al., Jets downstream of collisionless shocks, *Space Sci. Rev.*, 214(5), 81, doi:10.1007/s11214-018-0516-3, 2018.

Raptis, S., T. Karlsson, F. Plaschke, A. Kullen, and P.-A. Lindqvist, Classifying magnetosheath jets using MMS: Statistical properties, *Journal of Geophysical Research: Space Physics*, 125(11), e2019JA027,754, doi:<https://doi.org/10.1029/2019JA027754>, 2020.

Valek, P. W., et al., Hot flow anomaly observed at jupiter's bow shock, *Geophysical Research Letters*, 44(16), 8107–8112, doi:10.1002/2017GL073175, 2017.

Vuorinen, L., H. Hietala, A. T. LaMoury, and F. Plaschke, Solar wind parameters influencing magnetosheath jet formation: Low and high IMF cone angle regimes, *Journal of Geophysical Research: Space Physics*, 128(10), doi:10.1029/2023JA031494, 2023.

REVIEWERS' COMMENTS

Reviewer #2 (Remarks to the Author):

As the authors have addressed all my comments, I am glad to recommend the paper for publication.